# A network analysis revealed the essential and common downstream proteins related to inguinal hernia

Yimin Mao[1,2☯], Le Chen[1☯], Jianghua Li[1], Anna Junjie Shangguan[3], Stacy Kujawa[4], Hong Zhao[4]*

1 School of Information and Technology, Jiangxi University of Science and Technology, Jiangxi, China, 2 Applied Science Institute, Jiangxi University of Science and Technology, Jiangxi, China, 3 Department of Radiology, Feinberg School of Medicine, Northwestern University, Chicago, Illinois, United States of America, 4 Division of Reproductive Science in Medicine, Department of Obstetrics and Gynecology, Feinberg School of Medicine, Northwestern University, Chicago, Illinois, United States of America

☯ These authors contributed equally to this work.
* h-zhao@northwestern.edu

**Data Availability Statement:** All relevant data are within the manuscript and its Supporting Information files.

## Abstract

Although more than 1 in 4 men develop symptomatic inguinal hernia during their lifetime, the molecular mechanism behind inguinal hernia remains unknown. Here, we explored the protein-protein interaction network built on known inguinal hernia-causative genes to identify essential and common downstream proteins for inguinal hernia formation. We discovered that PIK3R1, PTPN11, TGFBR1, CDC42, SOS1, and KRAS were the most essential inguinal hernia-causative proteins and UBC, GRB2, CTNNB1, HSP90AA1, CBL, PLCG1, and CRK were listed as the most commonly-involved downstream proteins. In addition, the transmembrane receptor protein tyrosine kinase signaling pathway was the most frequently found inguinal hernia-related pathway. Our in silico approach was able to uncover a novel molecular mechanism underlying inguinal hernia formation by identifying inguinal hernia-related essential proteins and potential common downstream proteins of inguinal hernia-causative proteins.

## Introduction

In general surgery, inguinal hernia repair is one of the most routine operations worldwide. More than one in four men can expect to undergo inguinal hernia repair during their lifetime [1, 2]. Annual health care costs directly attributable to inguinal hernia exceed $2.5 billion in the US [3]. Inguinal hernias can be classified as either indirect, where the bowel herniates through a defective inguinal ring, or direct, where the bowel protrudes through the weakened lower abdominal muscle wall [3–6]. At times, inguinal hernias can cause severe complications, such as incarceration and strangulation and currently, surgery is the only treatment option for the management of inguinal hernia. Unfortunately, complications such as postoperative pain, nerve injury, infection, and recurrence continue to challenge surgeons and patients [5, 7–9]. Despite its prevalence in patients, the molecular mechanisms that predispose individuals to develop inguinal hernias are still unknown.

**Funding:** The authors received no specific funding for this work.

**Competing interests:** The authors have declared that no competing interests exist.

There are several key risk factors for inguinal hernias. For instance, men are more predisposed to developing inguinal hernias and have a lifetime risk of 27% compared with a 3% lifetime risk in women [1]. Old age is also a significant risk factor for hernia with incidence peaking between 60 and 75 years of age, with approximately 50% of men developing an inguinal hernia by the age of 75 [10–13]. The risk of inguinal hernia also increases among first-degree relatives of inguinal hernia patients, indicating a genetic risk factor for inguinal hernia development [14, 15]. Additionally, individuals with connective tissue genetic diseases such as cutis laxa [16], Marfan syndrome [17], and Ehlers-Danlos syndrome [18] have a greater risk of developing inguinal hernias. To date, only a small number of those candidate genes have been investigated [19–24]. Among those findings, a large genome-wide association study recently identified four novel inguinal hernia susceptibility loci in the regions of EFEMP1, WT1, EBF2, and ADAMTS6. Moreover, mouse connective tissue and network analyses showed that two of these genes (EFEMP1 and WT1) are critical for connective tissue maintenance/homoeostasis given their expression [21]. However, inguinal hernia-causative genes and their corresponding proteins in the pathophysiology of inguinal hernia are still unknown.

Recently, big data analysis has enabled the discovery of crucial disease-causative genes and pathogenic mechanisms by exploring publicly available Online Mendelian Inheritance in Man (OMIM) databases. Furthermore, the protein-protein interactions (PPIs) between corresponding proteins of disease-causative genes were studied by construction of the PPI network [25]. The PPI network for studying human diseases has achieved noteworthy results [26–30]. Several previous studies showed the feasibility of computational approaches to predict gene essentiality and morbidity [31–34]. For example, the topological properties of PPIs have been employed to identify essential proteins in various organisms [35, 36]. The main concept is the "centrality-lethality rule", in which highly connected hub proteins with a central role are more essential to survival in the PPI network [34]. Although there is still significant debate regarding this rule, several studies suggest a correlation between topological centrality and protein essentiality [27, 37, 38]. Additionally, there may be common downstream proteins, which maximally connect with those inguinal hernia-causative proteins through either direct or indirect interaction. We applied these concepts to calculate the essentiality of each protein in the inguinal hernia-PPI network to define crucial inguinal hernia-causative proteins and their downstream proteins.

In the present study, we constructed a PPI network based on inguinal hernia-causative genes imported from the OMIM database. We then identified key protein nodes of significant influence using topological network indices, namely, degree, betweenness, closeness, and eigenvector centrality. Our integration of network topological properties and protein cluster information revealed several highly ranked essential proteins related to inguinal hernia formation. We also performed the functional enrichment analysis of those essential proteins and identified several common downstream key proteins. Our results revealed the novel molecular mechanisms associated with human inguinal hernias which may serve as the potential drug targets to combat this prevalent disease.

## Materials and methods

### The analytical framework

To investigate the essential and common proteins related to inguinal hernia, the analytical framework is schematically illustrated in Fig 1. The whole process in this study consists of three main steps–construction, processing, and detecting: 1) Construction involves obtaining the inguinal hernia causal genes from the OMIM database and creating the inguinal hernia PPI networks by inputting the inguinal hernia-causative genes into Interologous Interaction

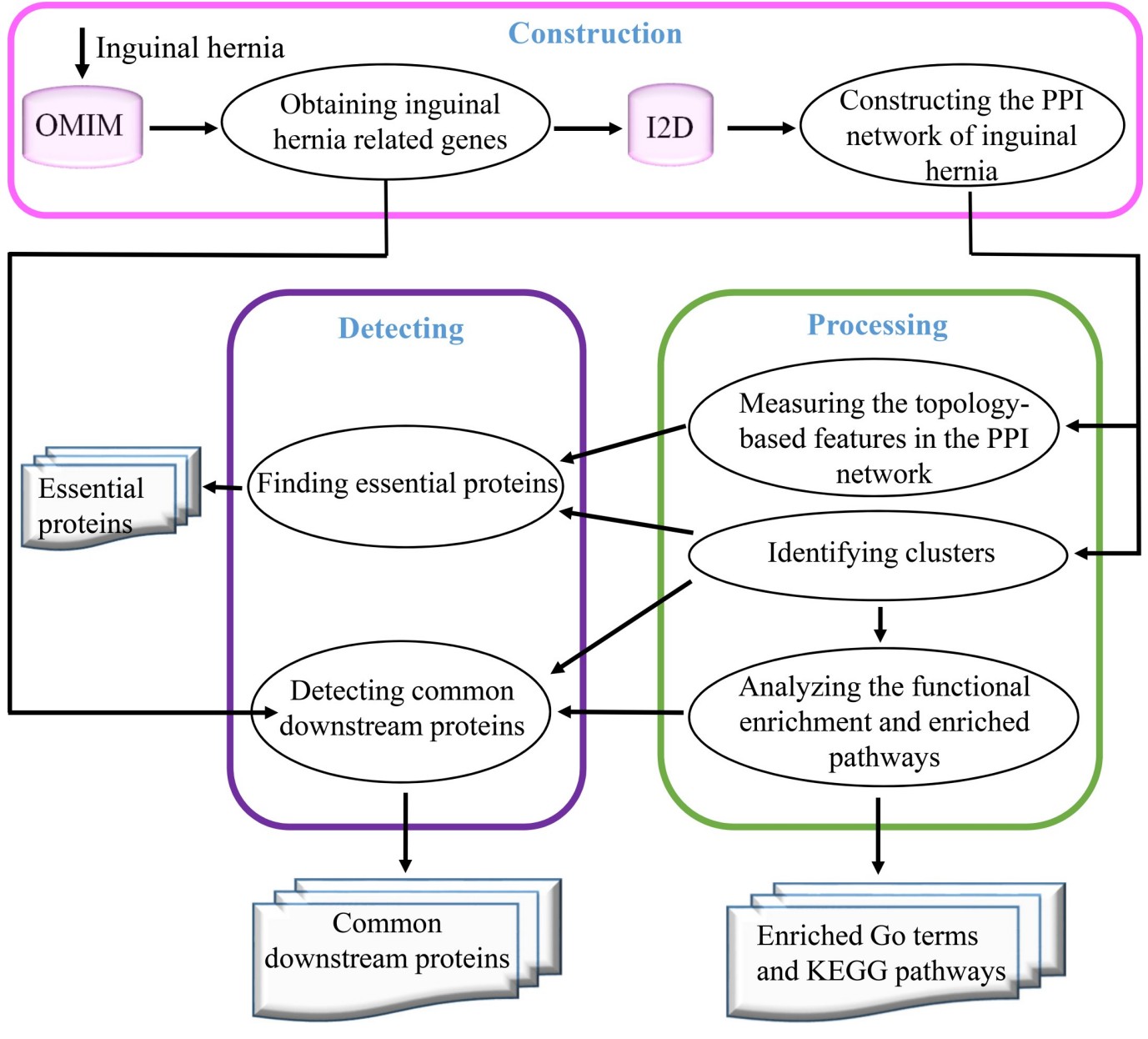

**Fig 1. The overall framework to detect essential and common downstream proteins.**

Database (I2D); 2) Processing involves measuring topology-based features in the protein interaction, identifying clusters, and analyzing the functional enrichment and pathways; and 3) Detecting involves defining essential and common downstream proteins.

## Database

Two main databases, OMIM and I2D, were used in this study. The OMIM database is a comprehensive research resource of curated descriptions of human genes and genetic disorders [39]. I2D is a comprehensive database integrating experimental and predicted PPIs which possesses 38 well-known human protein interaction databases (e.g., Inact, BINT, HRPD, and MINT) containing over 230,000 experimental and approximately 70,000 predicted PPIs from

human sources (http://ophid.utoronto.ca/i2d). Identified proteins were unified using the protein IDs defined in the Uniprot database [25]. The database versions of OMIM and I2D were updated in December 2017.

## Inguinal hernia-related genes and construction of PPI networks

Our input of the term "inguinal hernia" into the OMIM database yielded a list of hereditary genes of inguinal hernia from the OMIM morbid map (http://www.omim.org). Then, these gene names were submitted to the I2D database with human as the chosen target organism and the resultant PPIs were generated including predicted and experimental protein interactions. To increase the data reliability of protein interactions, all predicted homologous protein interactions were excluded. The remaining protein interactions were employed to construct the inguinal hernia PPI network in which proteins serve as nodes and protein-protein interactions serve as edges. Protein identifiers were unified using the protein IDs defined in the UniProt database [24]. Because some proteins were given multiple names, the results in tables and figures were presented in the format of gene names and UniProt IDs to avoid the ambiguous referring.

## Identifying clusters

To further understand the biological function of the PPI network generated by using inguinal hernia-causative genes, a clique percolation clustering method (CPM) [30], which is a partition algorithm, was first used to identify dense subgraphs with various $k$-cliques ($k$ is the size of a clique, a $k$-clique at $k = 3$ is equivalent to a triangle) and then explore overlapping clusters. A cluster is composed of a series of adjacent $k$-cliques, which can be reached from one another through the overlapping protein nodes. In the present study, the inguinal hernia PPIs were further analyzed using CFinder-2.0.6, an open source CPM software platform, for detecting the densely connected regions in the PPI network that have possible overlapping clustering between various $k$-cliques, leading to speculating their specific biological functions [29].

## Detecting essential proteins

Essential proteins exert vital roles in cellular processes and are indispensable for survival or reproduction [29, 40]. Essential proteins are also critical for the development of human diseases [41]. Here, the topological features of the PPI network were used to detect the role of essential proteins in inguinal hernia disease. These methods are based on the centrality-lethality rule, which means essential proteins tend to form hubs (highly connected protein nodes) in the PPI network. The removal of essential proteins causes the PPI network to break down [28]. Genome-wide studies show that deletion of a hub protein is more likely to be lethal than deletion of a non-hub protein [27]. A number of categories for defining centralities, such as degree centrality (DC) [27], closeness centrality (CC)[26], betweenness centrality (BC) [42, 43], and eigenvector centrality (EC) [44], have been proposed to characterize the inguinal hernia PPI network and the participating proteins for predicting essential proteins.

Suppose a PPI network is regarded as an undirected graph $G$ ($V$, $E$) with proteins as nodes ($V$) and interactions as edges ($E$), where $u$ represents a protein node in the PPI network and $v$ is any protein nodes other than $u$ in the network, four features of the inguinal hernia PPI network were characterized as follows:

(1) Degree centrality (DC) measures the number of interactions that a protein has. It can be defined as the following equation [27].

$$DC(u) = \sum_u edg(u, v), \tag{Eq 1}$$

where $edg(u,v)$ is the interaction between $u$ and $v$. if such interaction does exist, the $edg(u,v)$ is one. If not, it is zero.

(2) Closeness centrality (CC) is a measurement of how close a protein is to others. The CC of a protein node in the PPI network is considered as the reciprocal sum of its distance to all other nodes. It can be defined as the following equation [26].

$$CC(u) = \frac{N-1}{\sum\limits_{v \in V} dis(u,v)}, \qquad (Eq\ 2)$$

where $N$ is the number of the protein node in $V$ and $dis(u,v)$ is the distance between $u$ and $v$.

(3) Betweenness centrality (BC) measures the positional influence of a protein in the networks. The BC of a node $k$ in the PPI network is defined as the relative stress centrality that can quantify the extent to which node $k$ monitors the communication between other nodes. It can be defined as the following equation [42, 43].

$$\delta_{uv}(k) = \frac{p(u,k,v)}{p(u,v)}, u \neq k \neq v,$$
$$BC(k) = \sum_{u \in V} \sum_{v \in V} \delta_{uv}(k), \qquad (Eq\ 3)$$

where $\delta_{uv}(k)$ is the fraction of the shortest paths that pass through the node $k$ in the PPI network.

(4) Eigenvector centrality (EC) measures the relative number of interaction connecting one protein to its surrounding proteins. The EC of a protein node in the PPI network assumes that the centrality value of a protein node depends on the values of each adjacent node, which is defined as the following equation [44].

$$EC(u) = e_{\max}(u), \qquad (Eq\ 4)$$

where $e_{\max}$ denotes the principal eigenvector of the adjacency matrix (PPI network considered as a matrix) and $e_{\max}(u)$ denotes the $u$-$th$ component of the principal eigenvector.

Although the computation of centrality based on the network topology has become an important method for identifying essential proteins, it is difficult to identify many essential proteins that have low connectivity in the PPI network [45]. Recently, the majority of studies have shown that the essentiality of proteins has a strong correlation with clusters [46, 47], which indicates that essential proteins tend to gather in clusters. To further analyze the PPI network employing both topology features and the cluster characteristics, a novel edge clustering coefficient (ECC) algorithm was designed to better detect essential proteins [46].

First, the cluster centrality of a protein $i$, which means the overlapping cluster number of a protein, is defined as follows:

$$fun\_c(i) = \sum_{k=1}^{m}(|e(i,j)|),\ i,j \in V(C_k) \qquad (Eq\ 5)$$

where $fun\_c(i)$ is the cluster centrality of a protein $i$, $m$ is the number of clusters containing $i$, $j$ is any proteins other than $i$ in the PPI network, $e(i,j)$ is an edge between $i$ and j, $C_k$ is the $k^{th}$ cluster ($1 \leq k \leq m$), and $V(C_k)$ is the node set of $C_k$.

Next, together with cluster centrality, the PPI network topology features are added to measure the essential protein via the shape of the network [46]. Suppose the most appropriate topology feature was defined as $TopoCentrality$, to integrate $TopoCentrality(i)$ and cluster

centrality *fun_c*(*i*), the harmonic centrality (HC) of a protein *i* was defined as follows:

$$HC(i) = \delta * \frac{TopoCentrality(i)}{TopoCentrality_{max}} + (1 - \delta) * \frac{fun\_c(i)}{fun\_c_{max}}, \qquad (\text{Eq 6})$$

where *TopoCentrality*(*i*) is the most appropriate topology centrality of *i*, *TopoCentrality*$_{max}$ is the maximum value of *TopoCentrality*(*i*), *fun_c*$_{max}$ is the maximum value of *fun_c*(*i*), $\delta$ is a tunable factor in the range [0,1] which is used to adjust the weights of *TopoCentrality*(*i*) and *fun_c*(*i*). Generally, $\delta$ is set to 0.5.

## Gene ontology and pathway enrichment analyses

To further explore the biological roles of the genes in clusters, Gene Ontology (GO) term and Kyoto Encyclopedia of Genes and Genomes (KEGG) pathway enrichment analyses were conducted using the tools from the Database for Annotation, Visualization and Integrated Discovery (DAVID, Version 6.8), which is a web-based bioinformatics resource, an integrated analysis tool, and a biological knowledge base [48]. GO term enrichment and KEGG pathway analyses were performed using the GO knowledgebase (http://www.geneontology.org) and KEGG (http://www.genome.jp/kegg/) database, respectively.

## Finding common downstream proteins

To determine the common downstream proteins related to inguinal hernia, a novel deformation breadth-first search (DBFS) algorithm was designed. All causative proteins related to inguinal hernia in the PPI network were assumed as the destination set *D*. Proteins closely linked to destination set *D* were considered as the common downstream proteins because these proteins in the same clusters closely interact with each other to play a critical role in inguinal hernia development. The brief procedure for finding common downstream proteins are summarized here. Firstly, the DBFS algorithm found all adjacent proteins (i.e., one hop proteins) for every destination protein in set *D*. These one hop proteins, if they were not being visited before and were not included in the destination set *D*, were able to be visited. Then, all adjacent proteins (i.e., two hops proteins) of every one hop protein were searched and operated using the same rule. This step was repeated until the path length reached the constrain threshold. The detailed procedure of DBFS is shown as below:

Input: *G* = <*V*,*E*>,*G*∈ *inguinal hernia PPI networks*; *L* = *the constrain threshold of path length*;
 *D* = (*d*$_1$,*d*$_2$,...,*d*$_n$),*d*$_i$∈*destination protein set*.
InitQueue(*Q*); InitQueue(*T*); //set *Q* and *T* as queue
For (*i* = 0; *i*<*n*; *i*++) EnQueue(*Q*, *d*$_i$); //put each destination protein into queue *Q*
While (!QueneEmpty(*Q*)) // the queue *Q* of destination proteins is not empty
 For (*V* = 0; *V*<*G.VexNum*; *V*++) Visited [*V*] = *False*; //set all the proteins in *G* as unvisited label
 EnQueue(*T*, -1); //-1 regard as the label of each hop
 DeQueue(*Q*, *u*); *distance* = 1; // *u* is the node out of queue Q
 While (*distance*< = *L*)
 While (*u*! = -1)
 For (*w* = FirstAdjVex(*G*,*u*);*w*> = 0;*w* = NextAdj(*G*,*u*,*w*)) //obtain all adjacent nodes of *u*
 If ((!visited[*w*]) && (*w* not in *D*)) //the adjacent point w is not visited and not belongs to D
 Visited[*w*] = TRUE; Visited(*w*,*distance*);
 EnQueue(*T*,*w*);
 Endfor
 DeQueue(*T*, *u*);

```
            EndWhile
            EnQueue (T, -1); distance = distance+1; DeQueue(T,u);
        EndWhile
EndWhile
```

## Statistical analysis

To determine whether the value significantly deviates from the mean value, the modified Thompson tau technique was employed. Its basic concepts are as follows:

$$\tau = \frac{t_{\alpha/2} \cdot (n-1)}{\sqrt{n} \cdot \sqrt{n - 2 + {t_{\alpha/2}}^2}} \qquad \text{(Eq 7)}$$

where $n$ is the number of data points, $t_{\alpha/2}$ is the critical student's $t$ value, $SD$ is sample standard deviation, $\alpha = 0.05$, and df = $n$-2. $^*$ > $mean \pm (\tau \cdot SD)/2$, $^{**}$ > $mean \pm \tau \cdot SD$.

# Results

## Inguinal hernia-causative genes and the inguinal hernia PPI network construction

A total of 83 inguinal hernia-causative genes were obtained from the OMIM database as shown in Table 1. Based on known PPIs, the interactions of those gene products (inguinal hernia-causative proteins) were investigated by exploring the I2D database to build the inguinal hernia PPI network. To achieve this, all genes had to contain the loci with known encoding protein profiles in Uniprot. After removal of four genes without corresponding known coding proteins (i.e., SRS, DIH2, ICR1, and H19) from the inguinal hernia-causative gene list, 79 genes were used to explore the I2D database. Our input of inguinal hernia-related proteins into I2D yielded 8,215 interactions in inguinal hernia PPI networks. After removal of homologous predicted interactions, 4,201 interaction edges accompanied by 2,666 protein nodes were eventually utilized to construct the inguinal hernia PPI networks (Fig 2).

## Cluster analysis

To further determine inguinal hernia-causative proteins based on the overlapping clusters (cliques) in the inguinal hernia PPI networks, cluster analysis was conducted using the Cluster-One plug-in of the CFinder 2.0.6 software. The clusters with densely connected nodes in the inguinal hernia PPI network were detected. The numbers of clusters were 784, 245, and 62, which corresponded to the clique percolation parameter $k$ = 3, 4, and 5, respectively. A network diagram of clusters at $k$ = 4 was shown in Fig 3. The overlapping cluster numbers for a protein that participated in clusters are shown in Table 2. PIK3R1, PTPN11, SOS1, TGFBR1, TGFBR2, CDC42, KRAS, HRAS, RET, and PDGFRA were listed as the top ten proteins based on the overlapping cluster number of hernia-causative genes, in which PIK3R1 and PTPN11 were significantly involved in the inguinal hernia PPI network.

## Detecting essential proteins

Four topological features (i.e., DC, CC, BC, and EC) were calculated for identifying the essential proteins in the inguinal hernia PPI network. DC, CC, BC, and EC were weighted equally when calculating the essential proteins. The top 20 of those essential proteins were ranked and shown in Table 3. PIK3R1 (P27986), CDC42 (P60953), CTFR (P13569), TGFBR1 (P36897), and PTPN11 (Q06124) were the top five proteins with visibly higher DC. These results suggest

**Table 1. Inguinal hernia-causative genes from OMIM.**

| Uniprot ID | Gene | Uniprot ID | Gene | Uniprot ID | Gene | Uniprot ID | Gene |
|---|---|---|---|---|---|---|---|
| P42345 | MTOR | O75369 | FLNB | P53667 | LIMK1 | P49918 | CDKN1C |
| Q02809 | PLOD1 | Q7Z494 | NPHP3 | P08123 | COL1A2 | P19544 | WT1 |
| P98160 | HSPG2 | P58012 | FOXL2 | P13569 | CFTR | O95967 | EFEMP2 |
| P60953 | CDC42 | O00469 | PLOD2 | P43694 | GATA4 | P50454 | SERPINH1 |
| Q8TAD8 | SNIP1 | Q8NEZ3 | WDR19 | P13497 | BMP1 | O60706 | ABCC9 |
| Q04721 | NOTCH2 | P16234 | PDGFRA | P11362 | FGFR1 | P01116 | KRAS |
| P35354 | PTGS2 | Q86XX4 | FRAS1 | Q8WW38 | ZFPM2 | P02458 | COL2A1 |
| P00797 | REN | Q99697 | PITX2 | P07951 | TPM2 | P13647 | KRT5 |
| O95259 | KCNH1 | Q9NQX1 | PRDM5 | Q01974 | ROR2 | Q16671 | AMHR2 |
| P61812 | TGFB2 | P27986 | PIK3R1 | P37058 | HSD17B3 | Q06124 | PTPN11 |
| Q07889 | SOS1 | Q16637 | SMN1 | P36897 | TGFBR1 | Q5SZK8 | FREM2 |
| P22888 | LHCGR | P50443 | SLC26A2 | Q13285 | NR5A1 | Q9Y625 | GPC6 |
| Q12805 | EFEMP1 | A1X283 | SH3PXD2B | P20908 | COL5A1 | O95455 | TGDS |
| P02461 | COL3A1 | O95450 | ADAMTS2 | P52895 | AKR1C2 | P12644 | BMP4 |
| P05997 | COL5A2 | P35916 | FLT4 | P17516 | AKR1C4 | P10600 | TGFB3 |
| O14793 | MSTN | Q99519 | NEU1 | P07949 | RET | Q9UBX5 | FBLN5 |
| O00755 | WNT7A | P22105 | TNXB | P36894 | BMPR1A | – | SRS |
| P37173 | TGFBR2 | Q9ULC3 | RAB23 | P54886 | ALDH18A1 | – | DIH2 |
| P16278 | GLB1 | Q9NWM8 | FKBP14 | Q02962 | PAX2 | – | ICR1 |
| Q9BWF2 | TRAIP | Q8N8U9 | BMPER | P05093 | CYP17A1 | – | H19 |
| P41221 | WNT5A | P15502 | ELN | P01112 | HRAS | | |

– means no-Uniprot-ID

the importance and extensive involvement of these proteins in inguinal hernia pathogenesis. In addition, the top five proteins with higher CC were UBC (P0CG48), PIK3R1 (P27986), CDC42 (P60953), TGFBR1 (P36897), and PTPN11 (Q06124); with higher BC were PIK3R1 (P27986), UBC (P0CG48), CDC42 (P60953), CTFR (P13569), and TGFBR1 (P36897); and with higher EC were PIK3R1 (P27986), PTPN11 (Q06124), CDC42 (P60953), TGFBR1 (P36897), and SOS1 (Q07889). Consequently, PIK3R1, CDC42, and TGFBR1 were always listed in the top five proteins for all topological categories, and PTPN11 was listed in the top five proteins under three topological categories (DC, CC, and EC). UBC was listed as the first and second protein in CC and BC, respectively. CTFR was listed as the third and the fourth protein under two topological categories (DC and BC), and SOS1 was listed as the fifth protein in the topological category EC. Thus, proteins PIK3R1, CDC42, TGFBR1, PTPN11, UBC, CTFR, and SOS1 may play an important role in the inguinal hernia PPI network.

As expected, we observed different proteins present under various topological features, because each topological feature depicts only certain features of the PPI network and cannot include the entire topological information of the PPI network. Thus, we further identified the essential proteins in the PPI network using a comprehensive edge clustering coefficient (ECC) algorithm. The ECC method considers both the clustering characteristics and the topological features of a protein. Because the shape of the inguinal hernia PPI network is similar to a star topology as shown in Fig 1, DC is more suitable to find essential proteins. As a result, the topological feature of harmonic centrality (HC) calculating from the ECC algorithm was used to further define DC. The essential proteins ranked by HC are shown in Fig 4, in which PIK3R1, PTPN11, TGFBR1, CDC42, SOS1, and KRAS are shown as the significantly enriched top-ranking hub proteins in the inguinal hernia PPI network using the Thompson Tau test.

Together, our results show that PIK3R1, PTPN11, TGFBR1, CDC42, and SOS1 are probably the most essential proteins involved in human hernia formation.

## Gene ontology and pathway enrichment analysis

We performed functional enrichment analysis including GO term enrichment and KEGG pathway analysis on 784 clusters, k = 3 of the inguinal hernia PPI network. In GO term

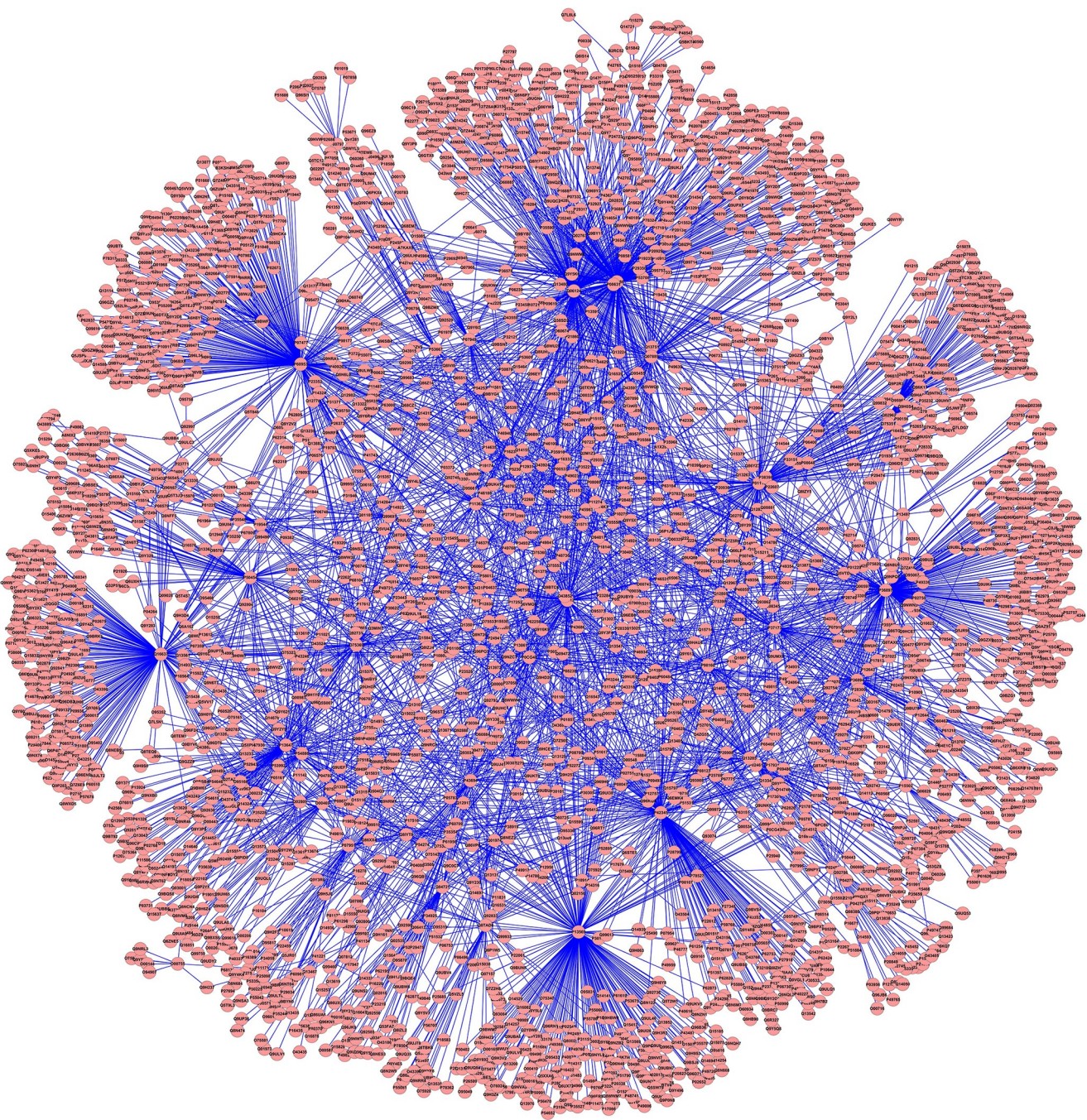

**Fig 2. PPI networks related to inguinal hernia with 4,201 interactions and 2,666 protein nodes.** Red circles represent protein nodes and blue lines indicate the protein-protein interactions.

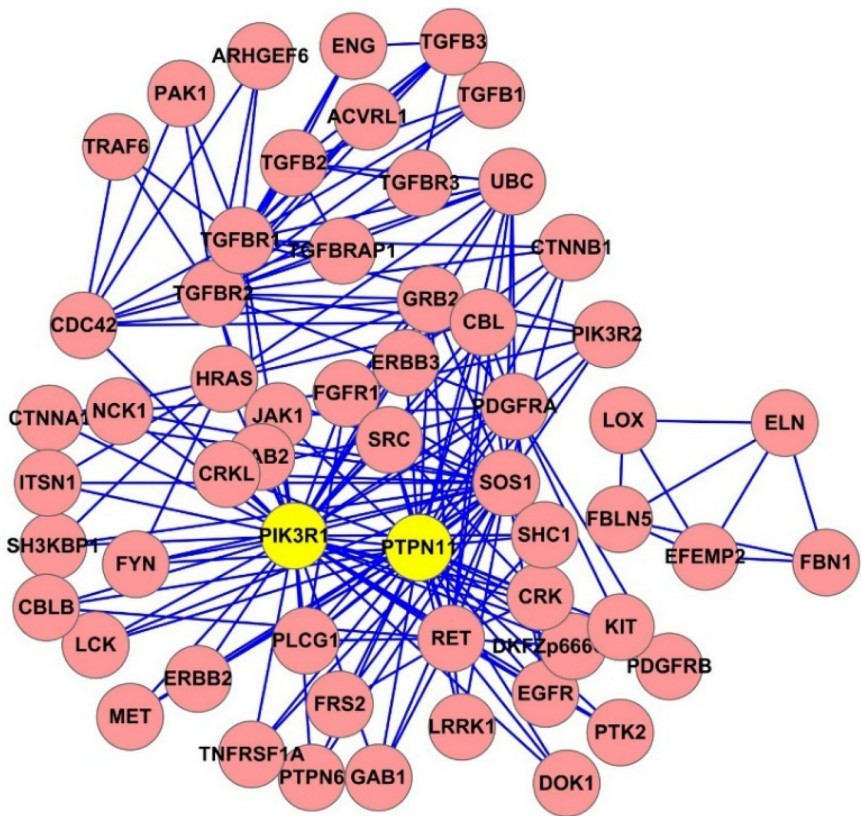

**Fig 3. The clusters of inguinal hernia-causative genes in the PPI network.** 245 clusters at k = 4. The yellow core clusters are defined by significant involvement ranking calculated in Table 2 using the Thompson Tau test.

analysis, three categories were used including biological processes, cellular components, and molecular function. The top ten GO terms in three categories are described in Table 4 with $p$ values < 0.01. The top seven significant terms in the biological processes category were peptidyl-tyrosine phosphorylation, transmembrane receptor protein tyrosine kinase (RTK) signaling pathway, vascular endothelial growth factor (VEGF) receptor signaling pathway,

**Table 2. Top 20 inguinal hernia-causative proteins based on the number of overlapping clusters.**

| Rank | Uniprot | Protein | Cluster # | Rank | Uniprot | Protein | Cluster # |
|------|---------|---------|-----------|------|---------|---------|-----------|
| 1 | P27986 | PIK3R1 | 207** | 11 | P11362 | FGFR1 | 23 |
| 2 | Q06124 | PTPN11 | 127* | 12 | P0CG48 | UBC | 19 |
| 3 | Q07889 | SOS1 | 101 | 13 | O75369 | FLNB | 16 |
| 4 | P36897 | TGFBR1 | 68 | 14 | Q13285 | NR5A1 | 14 |
| 5 | P37173 | TGFBR2 | 56 | 15 | P43694 | GATA4 | 11 |
| 6 | P60953 | CDC42 | 50 | 16 | P61812 | TGFB2 | 11 |
| 7 | P01116 | KRAS | 40 | 17 | P62993 | GRB2 | 11 |
| 8 | P11112 | HRAS | 37 | 18 | P36894 | BMPR1A | 9 |
| 9 | P07949 | RET | 26 | 19 | P10600 | TGFB3 | 8 |
| 10 | P16234 | PDGFRA | 23 | 20 | P35916 | FLT4 | 8 |

* $> mean \pm (\tau \cdot SD)/2$

** $> mean \pm \tau \cdot SD$

**Table 3. Top 20 inguinal hernia-causative proteins ranked by DC, CC, BC, and EC.**

| Rank | Uniprot | DC | Uniprot | CC | Uniprot | BC | Uniprot | EC |
|------|---------|-----|---------|--------|---------|------------|---------|--------|
| 1 | P27986 | 345 | P0CG48 | 0.4685 | P27986 | 1419876.10 | P27986 | 0.4584 |
| 2 | P60953 | 294 | P27986 | 0.4210 | P0CG48 | 1182660.50 | Q06124 | 0.2933 |
| 3 | P13569 | 264 | P60953 | 0.3880 | P60953 | 1163367.80 | P60953 | 0.2296 |
| 4 | P36897 | 264 | P36897 | 0.3867 | P13569 | 1051267.10 | P36897 | 0.1914 |
| 5 | Q06124 | 241 | Q06124 | 0.3778 | P36897 | 1025756.25 | Q07889 | 0.1813 |
| 6 | Q16637 | 226 | P37173 | 0.3737 | Q16637 | 906896.44 | P01116 | 0.1387 |
| 7 | P42345 | 191 | Q07889 | 0.3732 | Q06124 | 789004.06 | P01112 | 0.1352 |
| 8 | P01116 | 173 | Q13501 | 0.3689 | P42345 | 636229.44 | P0CG48 | 0.1323 |
| 9 | P01112 | 138 | P01112 | 0.3632 | P01116 | 589670.75 | P37173 | 0.1093 |
| 10 | P13647 | 126 | P01116 | 0.3617 | P01112 | 426118.50 | P07949 | 0.0993 |
| 11 | Q07889 | 126 | P62993 | 0.3617 | P13647 | 395877.22 | P13569 | 0.0989 |
| 12 | P50454 | 96 | P16234 | 0.3560 | Q07889 | 322209.06 | P42345 | 0.0963 |
| 13 | O75369 | 88 | P07949 | 0.3552 | P50454 | 301589.00 | P62993 | 0.0943 |
| 14 | P37173 | 87 | P22681 | 0.3549 | O95967 | 278135.72 | P11362 | 0.0859 |
| 15 | Q13285 | 69 | P11362 | 0.3541 | O75369 | 269789.28 | O75369 | 0.0835 |
| 16 | O95967 | 68 | P13569 | 0.3538 | Q13285 | 231566.27 | P16234 | 0.0804 |
| 17 | P11362 | 64 | O75369 | 0.3524 | P37173 | 212951.39 | P22681 | 0.0756 |
| 18 | P36894 | 62 | Q16637 | 0.3490 | Q8TAD8 | 178868.39 | P35222 | 0.0693 |
| 19 | P54886 | 58 | Q13285 | 0.3485 | P11362 | 172819.90 | Q16637 | 0.0691 |
| 20 | Q8TAD8 | 56 | P49841 | 0.3482 | P36894 | 156695.28 | P12931 | 0.0666 |

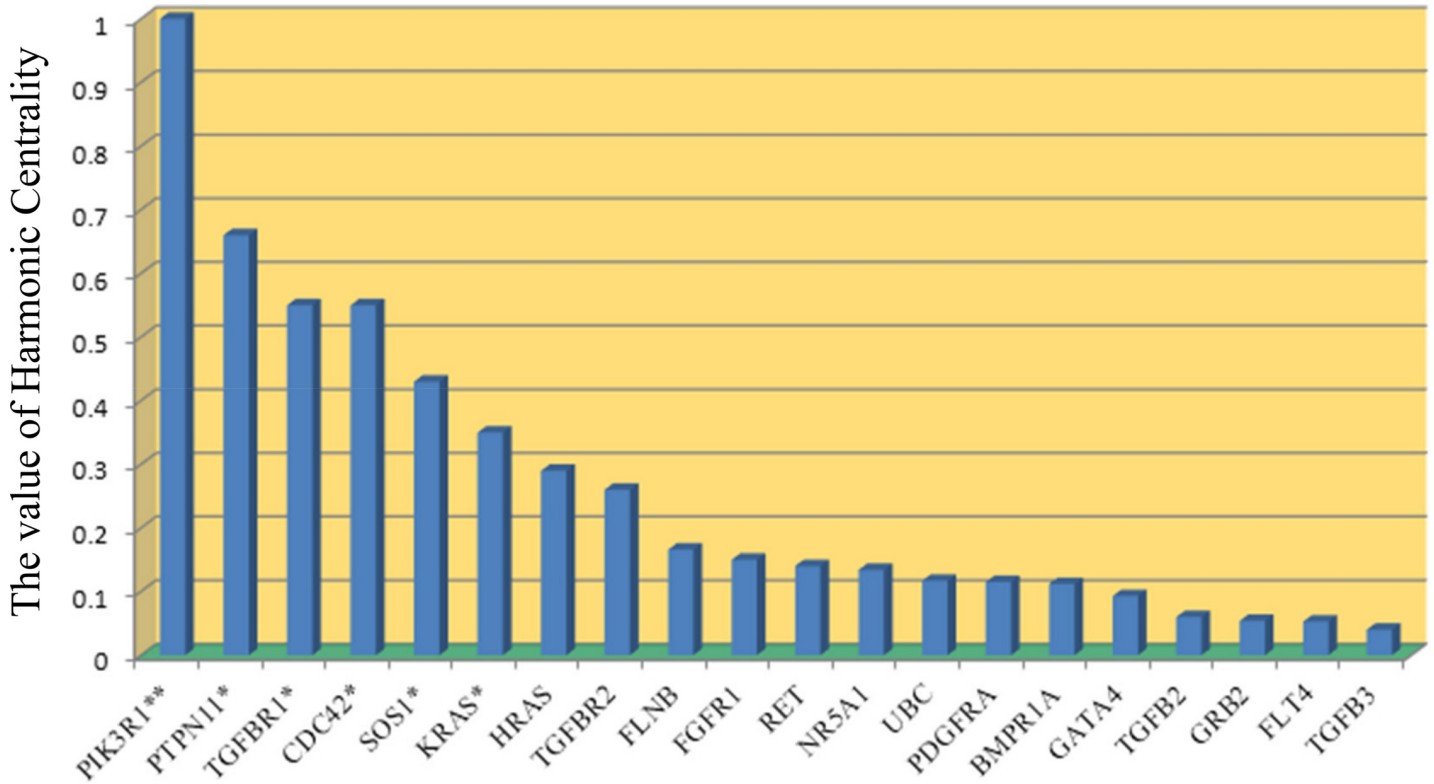

**Fig 4. Inguinal hernia-related essential proteins ranked by HC.** $^* > mean \pm (\tau \cdot SD)/2$, $^{**} > mean \pm \tau \cdot SD$.

**Table 4. The most significantly enriched GO terms.**

| Category | Terms | P-value |
|---|---|---|
| Biological Process | peptidyl-tyrosine phosphorylation | 2.89E-41 |
| | transmembrane receptor protein tyrosine kinase signaling pathway | 1.64E-32 |
| | vascular endothelial growth factor receptor signaling pathway | 8.75E-31 |
| | transforming growth factor beta receptor signaling pathway | 3.33E-30 |
| | signal transduction | 2.35E-29 |
| | MAPK cascade | 2.77E-29 |
| | regulation of phosphatidylinositol 3-kinase signaling | 4.29E-28 |
| | phosphatidylinositol-mediated signaling | 9.76E-27 |
| | epidermal growth factor receptor signaling pathway | 3.87E-26 |
| | positive regulation of GTPase activity | 1.39E-25 |
| Cellular Component | cytosol | 4.96E-37 |
| | plasma membrane | 5.80E-32 |
| | focal adhesion | 3.01E-29 |
| | cell-cell junction | 3.82E-20 |
| | receptor complex | 5.36E-19 |
| | membrane raft | 5.48E-17 |
| | extrinsic component of cytoplasmic side of plasma membrane | 5.49E-16 |
| | cytoplasm | 1.07E-14 |
| | cell-cell adherens junction | 1.49E-14 |
| | cell surface | 1.81E-13 |
| Molecular Function | protein binding | 6.13E-56 |
| | protein tyrosine kinase activity | 6.68E-40 |
| | transmembrane receptor protein tyrosine kinase activity | 2.11E-29 |
| | phosphatidylinositol-4,5-bisphosphate 3-kinase activity | 4.31E-28 |
| | protein kinase binding | 1.07E-24 |
| | Ras guanyl-nucleotide exchange factor activity | 8.92E-22 |
| | protein phosphatase binding | 1.03E-21 |
| | ATP binding | 8.75E-21 |
| | phosphatidylinositol-3-kinase activity | 3.34E-16 |
| | phosphatidylinositol 3-kinase binding | 2.85E-15 |

transforming growth factor beta (TGFβ) receptor signaling pathway, signal transduction, MAPK cascade, and regulation of phosphatidylinositol 3-kinase (PI3K) signaling. The Jak-STAT signaling pathway, insulin signaling pathway, fibroblast growth factor (FGF) receptor signaling pathway, and estrogen signaling pathway were also significantly enriched in this category of the GO term analysis (S1 Table). The most significant term was cytosol (p = 4.96E-37) in the cellular component category. The top five significant terms in the molecular function category were protein binding, protein tyrosine kinase activity, transmembrane RTK activity, phosphatidylinositol-4,5-bisphosphate 3-kinase activity, and protein kinase binding. These results show that the signaling pathways of growth factors, insulin, estrogen, transmembrane RTK, MAPK, and PI3K may play a vital role in the pathogenesis of inguinal hernia.

The KEGG pathway enrichment analysis revealed the significantly enriched genes in the enriched pathways (S1 Table). The top 15 enriched pathways are shown in Table 5. The most significantly changed pathway was proteoglycans in cancer. The next two pathways were pathways in cancer and the ErbB signaling pathway. In addition, the PI3K-Akt signaling pathway, MAPK signaling pathway, insulin signaling pathway, VEGF signaling pathway, TGFβ signaling pathway, Jak-STAT signaling pathway, and estrogen signaling pathway were listed as the

**Table 5. Most significantly enriched pathways determined by the KEGG pathway enrichment analysis and the involvement of top five essential proteins in the top 15 enriched pathways.**

| Terms | P-value | Top five essential proteins | | | | |
|---|---|---|---|---|---|---|
| | | PIK3R1 | PTPN11 | TGFBR1 | CDC42 | SOS1 |
| Proteoglycans in cancer | 1.12E-41 | + | + | - | - | + |
| Pathways in cancer | 5.98E-41 | + | - | + | + | + |
| ErbB signaling pathway | 9.34E-33 | + | - | - | - | - |
| Focal adhesion | 9.72E-32 | + | - | - | + | + |
| Ras signaling pathway | 1.00E-31 | + | + | - | + | + |
| Rap1 signaling pathway | 2.68E-31 | + | - | - | + | - |
| Chronic myeloid leukemia | 1.37E-30 | + | + | + | - | + |
| Pancreatic cancer | 1.56E-26 | + | - | + | + | - |
| Neurotrophin signaling pathway | 1.35E-25 | + | + | - | + | + |
| Adherens junction | 7.97E-24 | - | - | + | + | - |
| Renal cell carcinoma | 9.88E-24 | + | + | - | + | + |
| Fc epsilon RI signaling pathway | 4.25E-23 | + | - | - | - | + |
| Regulation of actin cytoskeleton | 4.57E-23 | + | - | - | + | + |
| FoxO signaling pathway | 1.18E-22 | + | - | + | - | + |
| PI3K-Akt signaling pathway | 2.54E-22 | + | - | - | - | + |

15th, 26th, 34th, 36th, 40th, 48th, and 50th significantly enriched pathways, respectively (S1 Table). All of these pathways are related to cell growth and proliferation. Among the top 15 enriched pathways, essential proteins PIK3R1, PTPN11, TGFBR1, CDC42, and SOS1 were involved in 14, 5, 5, 9, and 11 pathways, respectively (Table 5).

## Common downstream proteins

To investigate how these functionally diverse pathogenic proteins led to inguinal hernia formation, we further analyzed inguinal hernia PPI profiles using the DBFS algorithm to identify common downstream proteins of 79 inguinal hernia-causative proteins (Table 1). The top 21 common downstream proteins are shown in Table 6, in which UBC, GRB2, CTNNB1, HSP90AA1, PLCG1, CBL, and CRK were listed as the top 7 common downstream proteins

**Table 6. Top 21 common downstream proteins defined by the number of direct interactions with inguinal hernia-causative proteins.**

| Rank | Uniprot | Protein | Interacting # | Rank | Uniprot | Protein | Interacting # |
|---|---|---|---|---|---|---|---|
| 1 | P0CG48 | UBC | 50** | 11 | O00459 | PIK3R2 | 8 |
| 2 | P62993 | GRB2 | 19* | 12 | P00533 | EGFR | 8 |
| 3 | P35222 | CTNNB1 | 12* | 13 | P05067 | APP | 8 |
| 4 | P07900 | HSP90AA1 | 10 | 14 | P45983 | MAPK8 | 8 |
| 5 | P19174 | PLCG1 | 10 | 15 | P84022 | SMAD3 | 8 |
| 6 | P22681 | CBL | 10 | 16 | Q13547 | HDAC1 | 8 |
| 7 | P46108 | CRK | 10 | 17 | P01137 | TGFB1 | 7 |
| 8 | P12931 | SRC | 9 | 18 | P12830 | CDH1 | 7 |
| 9 | P29353 | SHC1 | 9 | 19 | P40763 | STAT3 | 7 |
| 10 | P49841 | GSK3B | 9 | 20 | Q03135 | CAV1 | 7 |
| | | | | 21 | Q15796 | SMAD2 | 7 |

$^* > mean \pm (\tau \cdot SD)/2$

$^{**} > mean \pm \tau \cdot SD$

(Fig 5). Most importantly, UBC, GRB2, and CTNNB1 were significantly enriched in the inguinal hernia PPI network. (Table 6).

## Discussion

Inguinal hernia is a multifactorial disease caused by endogenous factors including age, gender, anatomic variations, and inheritance as well as exogenous factors such as smoking, comorbidity, and outcomes from surgery [14]. Recently, we found that the conversion of testosterone to estradiol by the aromatase enzyme in lower abdominal muscle tissue in a humanized aromatase transgenic mouse model activates pathways for fibroblast proliferation and fibrosis, leading to intense lower abdominal muscle fibrosis, muscle atrophy, and inguinal hernia; fortunately, an aromatase inhibitor entirely prevents this phenotype [49]. In the present study, we explored the inherited aspects of inguinal hernia formation via data mining of inguinal hernia-causative genes exported from the OMIM database. Five essential proteins (PIK3R1, PTPN11, TGFBR1, CDC42, and SOS1) and three downstream common proteins (UBC, GRB2, and CTNNB1) were found to be related to inguinal hernia development. We also found that the signaling pathways of growth factors, transmembrane RTK, MAPK, and PI3K are highly associated with inguinal hernia disease. Furthermore, this data mining technique can be utilized for the analysis of the PPI networks of various human diseases to identify critical essential proteins contributing to human diseases.

The RTK pathways were shown in the biological process category of enriched GO terms including signaling pathways for VEGF receptor, TGFβ receptor, epidermal growth factor receptor, and insulin. The common downstream signaling of the RTK pathways such as the MAPK cascade and PI3K signaling was also found in the GO biological process analysis [50]. Furthermore, these growth factor-mediated RTK pathways were listed in the top 40 significantly enriched pathway via the KEGG pathway enrichment analysis. The GO term analysis also indicated that protein tyrosine kinase activity, transmembrane RTK activity, phosphatidylinositol-4,5-bisphosphate 3-kinase activity, as well as binding and activity of PI3K were related to human inguinal hernia diseases. Additionally, others also show that inguinal hernia-related essential proteins, PTPN11, CDC42, and SOS1 regulate the RAS/MAPK signaling pathway, which is another downstream effector of RTKs [51–54]. Furthermore, essential proteins, PIK3R1 and TGFBR1 are involved in the regulation of both the PI3K/AKT and RAS/MAPK signaling pathways [55–57]. Together, all of these analyses suggest that the RTK pathways such as the PI3K and MAPK pathways may play a critical role in the development of inguinal hernias.

To further reveal the genetic mechanism behind inguinal hernia, the DBFS algorithm was used to detect a series of common downstream proteins that directly interacted with inguinal hernia-causative proteins, in which UBC, GRB2, CTNNB1, HSP90AA1, PLCG1, CBL, and CRK are listed as the top seven downstream common proteins. Ubiquitin C, encoded by the UBC gene, maintains the cellular ubiquitin levels during stress. Protein ubiquitylation plays a key role in the regulation of multiple cellular events including the recognition of interacting proteins [58]. It is not surprising that UBC was highly positioned on the list of the downstream proteins related to inguinal hernia-causative proteins. It was also the highest ranked protein measured by CC and BC and may not have many direct neighbors as measured by DC. According to the previous study [28], if a node had low DC and high BC and EC, it would locate "centrality", where the exchange and transition of multitudes of data and resources allow the centrality node to arrive prior to the other nodes in the PPI network because of its short-distance path. Thus, UBC is likely to be adjacent to numerous inguinal hernia causative proteins. Furthermore, Akt ubiquitination plays an important role in the activation of Akt

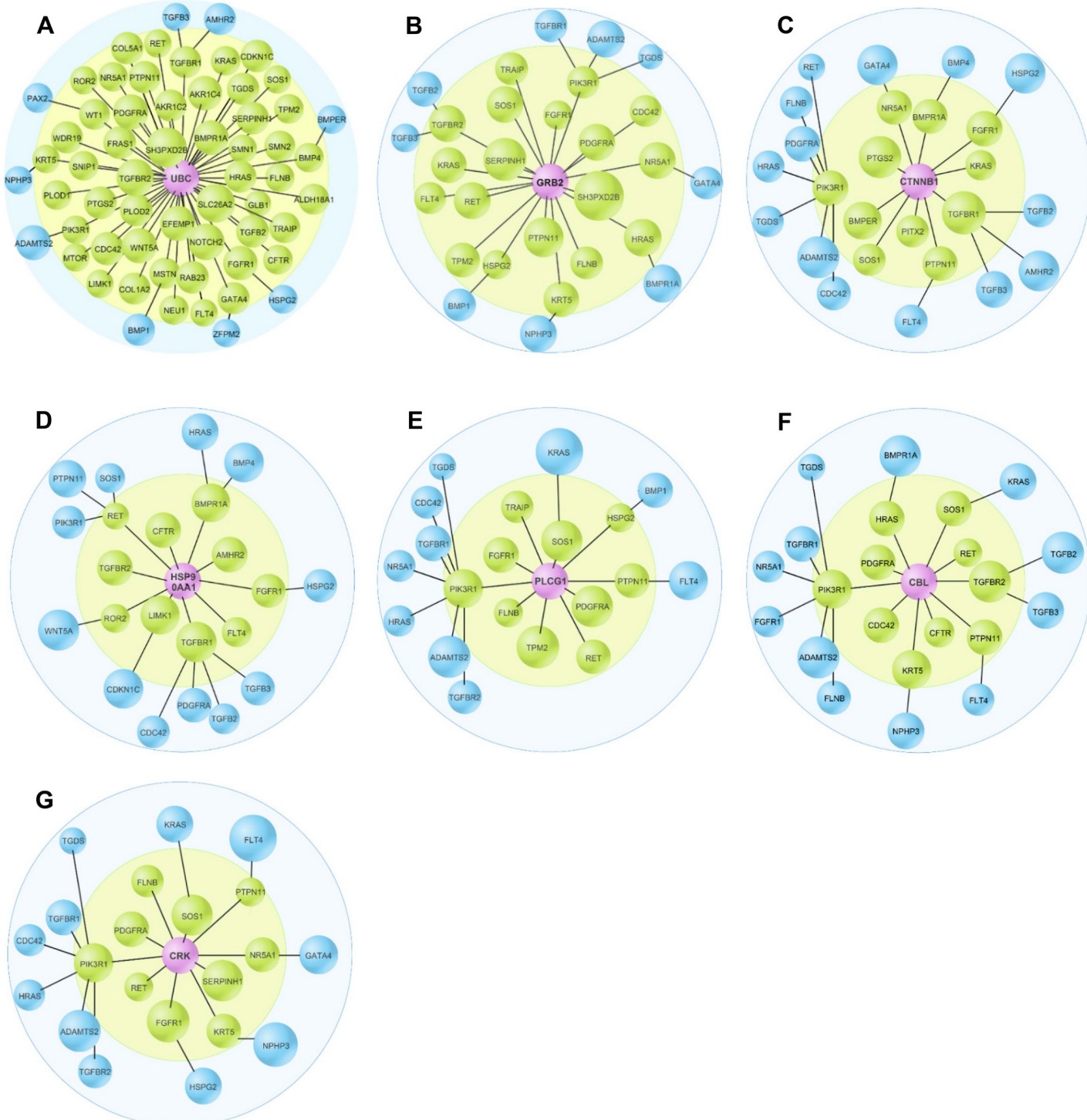

**Fig 5. The detailed interactive profiles of top 7 common downstream protein of inguinal hernia-causative proteins.** These common downstream proteins, UBC (A), GRB2 (B), CTNNB1 (C), HSP90AA1 (D), PLCG1 (E), CBL (F), and CRK (G), are highlighted in purple. The green proteins directly interacted with purple common downstream proteins. Blue proteins are the secondary contact proteins to purple common downstream proteins. The shortest distance between blue proteins and purple protein is 2.

signaling [59]. GRB2 (Growth factor receptor-bound protein 2) connects RTKs to RAS, leading to the activation of the RAS/MAPK pathway [60]. Another common downstream protein, CTNNB1 (Beta-catenin) is an essential part of the Wnt signaling pathway and is regulated by the PI3K/Akt pathways [61]. HSP90AA1 (heat shock protein 90 alpha family class A member 1) promotes autophagy and inhibits apoptosis through the PI3K/Akt/mTOR pathway [62]. PLCG1 (Phospholipase C, gamma 1) can be activated by RTKs [63]. CBL is an E3 ubiquitin-protein ligase involved in the formation of a covalent bond between ubiquitin and RTKs, leading to RTK protein ubiquitination and downregulation [64]. CRK is an adapter protein that binds to several tyrosine-phosphorylated proteins [65]. As discussed above, all of the top seven common downstream proteins are linked to the RTK pathways.

A major downstream mediator of RTKs, PI3K is a family of related intracellular signal transducer enzymes involved in cell growth, proliferation, differentiation, motility, survival, and intracellular trafficking [66]. It can be stimulated by diverse oncogenes and growth factor receptors such as receptors for insulin, insulin-like growth factor 1, TGFβ, VEGF, and platelet-derived growth factor. Estrogen can also up-regulate PI3K signaling [67]. The PI3K family is divided into three different classes (I, II, and III)[68]. Class IA PI3K is a heterodimeric enzyme containing a p85 regulatory and a p110 catalytic subunit. In the basal state, the interaction between p85 and p110 stabilizes and inhibits p110 catalytic activity. Upon activation by growth factors or other signals, ligand binding to RTKs promotes receptor activation. The p85 subunit binds to activated RTKs, recruiting p110 to the plasma membrane, where a conformational change induced by binding relieves inhibition of p110 catalytic activity. Three genes, PIK3R1, PIK3R2, and PIK3R3, encode the p85α, p85β, and p55γ isoforms of the p85 regulatory subunit of PI3K, respectively. The PIK3R1 gene also gives rise to two shorter isoforms, p55α and p50α, through alternative transcription-initiation sites [50, 66]. Interestingly, we found that PIK3R1 was listed as the first essential protein in relation to inguinal hernia disease. A previous study demonstrated that heterozygous mutations of PIK3R1(R649W) and the resultant impairment of the PI3K activation have been identified in patients with SHORT syndrome—a disorder characterized by Short stature, Hyperextensibility of joints and/or inguinal hernia, Ocular depression, Reiger anomaly and Teething delay [69]. This finding further emphasizes that PIK3R1 in the PI3K pathway is closely associated with the development of inguinal hernia.

Inguinal hernia formation is associated with increased lower abdominal muscle tissue fibrosis and muscle atrophy [49]. The exact role of these essential and common downstream proteins and the related signaling pathway in fibroblasts and myocytes of lower abdominal muscle tissue is unknown. Future studies will reveal the underlying molecular mechanisms for these proteins and pathways in fibroblast proliferation and fibrosis, myocyte function, and hernia formation and further target these essential and common downstream proteins for developing novel pharmacological approaches for preventing and treating recurrent inguinal hernia in high risk individuals. Our data along with previous findings regarding the effect of inhibitory mutation of PIK3R1 on SHORT syndrome-associated inguinal hernia indicate that the PI3K pathway, especially the essential protein, PIK3R1 is necessary for inguinal hernia development. A previous study showed that direct targeting of PIK3R1 in hepatic stellate cells inhibits liver fibrosis, indicating that PIK3R1 is probably participating in hernia-associated fibrosis [70]. In addition, overexpression of *pik3r1* in rat myotubes reduced insulin-stimulated PI3K/AKT activation [71], and *pik3r1* overexpression in mice decreased skeletal muscle insulin signaling [72]. Mice lacking both *pik3r1* and *pik3r2* in skeletal muscles exhibited severely impaired PI3K signaling in those muscles [73]. These animals showed reduced myocyte size and insulin-resistance in their skeletal muscles, demonstrating that *in vivo* class IA PI3K is both a vital regulator of muscle growth and a critical mediator of insulin signaling in the muscle. With these findings, we are planning to selectively delete *PIK3R1* in fibroblasts and/or myocytes to define the

relative roles of *PIK3R1* in fibroblasts and myocytes for maintaining abdominal muscle function and in pathologic processes such as fibrosis, atrophy, and hernia formation.

In summary, the present study deeply analyzed the protein-protein interaction on known inguinal hernia-causative genes from the OMIM database. Several essential proteins and common downstream proteins related to inguinal hernia diseases have been identified. The downstream signaling pathways of activated RTKs have been found to be highly associated with inguinal hernias. In the future, we will further determine how these essential proteins and the RTK signaling pathways such as the PI3K/Atk pathway contribute to the pathogenesis of the inguinal hernia formation.

## Supporting information

**S1 Table. The GO term and KEGG pathway enrichment analysis on the 784 clusters, k = 3 of the inguinal hernia PPI network.**
(XLSX)

## Author Contributions

**Conceptualization:** Yimin Mao, Hong Zhao.

**Data curation:** Yimin Mao, Le Chen.

**Formal analysis:** Yimin Mao.

**Funding acquisition:** Hong Zhao.

**Investigation:** Yimin Mao, Hong Zhao.

**Methodology:** Yimin Mao.

**Project administration:** Hong Zhao.

**Software:** Le Chen, Jianghua Li.

**Supervision:** Hong Zhao.

**Validation:** Yimin Mao, Hong Zhao.

**Visualization:** Yimin Mao, Le Chen.

**Writing – original draft:** Yimin Mao, Hong Zhao.

**Writing – review & editing:** Anna Junjie Shangguan, Stacy Kujawa.

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
