## [Decision Letter · Decision Letter 0]

18 Oct 2019

PONE-D-19-22307

A network analysis revealed the essential and common downstream proteins related to inguinal hernia

PLOS ONE

Dear Dr Zhao,

Thank you for submitting your manuscript to PLOS ONE. After careful consideration, we feel that it has merit but does not fully meet PLOS ONE’s publication criteria as it currently stands. Therefore, we invite you to submit a revised version of the manuscript that addresses the points raised during the review process.

The article is potentially interesting but some issues should be fixed before we can reconsider ti.

We would appreciate receiving your revised manuscript by Dec 02 2019 11:59PM. To enhance the reproducibility of your results, we recommend that if applicable you deposit your laboratory protocols in protocols.io, where a protocol can be assigned its own identifier (DOI) such that it can be cited independently in the future. For instructions see: http://journals.plos.org/plosone/s/submission-guidelines#loc-laboratory-protocols

We look forward to receiving your revised manuscript.

Kind regards,

Prof. Raffaele Serra, M.D., Ph.D

Academic Editor

PLOS ONE

Journal Requirements:

Additional Editor Comments:

This paper is potentially interesting for our journal, but I have some suggestions in order to improve the manuscript.

1. English language in your submission needs revision for style. It is a little hard to read. I fully understand how hard this can be, especially if English is not your first language. However, for readers to fully understand and benefit from your work it is crucial that the use of English is to a very high standard.

2. Your Discussion Section is too superficial in the current format. You have to discuss deeply your results, and in a more critical way. In particular, can you add a future perspectives comment based on your results?

Reviewers' comments:

Reviewer's Responses to Questions

**Comments to the Author**

1. Is the manuscript technically sound, and do the data support the conclusions?

Reviewer #1: Yes

2. Has the statistical analysis been performed appropriately and rigorously? 

Reviewer #1: Yes

3. Have the authors made all data underlying the findings in their manuscript fully available?

Reviewer #1: Yes

4. Is the manuscript presented in an intelligible fashion and written in standard English?

Reviewer #1: Yes

5. Review Comments to the Author

Reviewer #1: This is an interesting and important manuscript that adds significantly to the field and will have a noticeable impact. The manuscript is well-written and concise. The authors describe an interesting, well-thought and thoroughly designed approach for generation and analysis of PPI network associated with inguinal hernia. Using this technique, they were able not only to generate and analyze a PPI network containing proteins encoded by the inguinal hernia causative genes, but also to identify and characterize the essential and common downstream proteins related to the pathogenesis of this malady. What is even more important, this approach can be utilized for the analysis of the PPI networks associated with various human diseases. In other words, this work reports a crucial blueprint that will definitely simplify future research on bioinformatics-based identification of important players of various diseases. Furthermore, identified in this study essential and common downstream proteins related to inguinal hernia can serve as potential targets in the development of future therapeutics.

6. PLOS authors have the option to publish the peer review history of their article (what does this mean?). If published, this will include your full peer review and any attached files.

Reviewer #1: No

---

## [Author Response · Author response to Decision Letter 0]

2 Dec 2019

November 30, 2019

Academic Editor:

Prof. Raffaele Serra, M.D., Ph.D

Re: Manuscript “A network analysis revealed the essential and common downstream proteins related to inguinal hernia" (Manuscript #: PONE-D-19-22307)

Dear Dr. Serra:

Thank you very much for the prompt review of our manuscript and for your extremely helpful comments. In response to the comments from the reviewer and the editor, we have significantly revised the manuscript entitled, “A network analysis revealed the essential and common downstream proteins related to inguinal hernia”. 

We revised the manuscript in response to the comments from one reviewer and the editor. Thus, we respectfully request that the same editor and reviewer evaluate this revised version.

To fully address all of the comments, we have completely revised this manuscript including improvement of the discussion, fulfillment of PLOS ONE's style requirements, and English editing to meet the journal’s standard. The modified text is summarized and attached to this letter. We have revised discussion including addition of a paragraph to discuss the clinical significance and future direction of our findings and reorganization of the whole discussion. We have also revised the manuscript format to meet PLOS ONE's style requirements and English language of this manuscript to meet the language standards of PLOS ONE. We have included captions for our Supporting Information files at the end of our manuscript. The point-by-point response to the comments of the reviewer and the editor is attached to this cover letter. Changes to the text as a result of the new information and the reviewers’ comments are highlighted in blue in the manuscript and the corresponding page and line numbers regarding the highlighted text will be found in the “Revised Manuscript with Track Changes” file.

We appreciate the critiques and the opportunity to submit the revised manuscript. We also look forward to a favorable decision on publication of the revised manuscript in PLOS ONE.

Best Regards,

Hong Zhao, M.D., Ph.D.

Research Associate Professor

Department of Obstetrics and Gynecology

Robert H Lurie Comprehensive Cancer Center

Feinberg School of Medicine at Northwestern University

303 E. Superior Street, Suite 04-121

Chicago, Illinois 60611

Tel: 312-503-0780

Email: h-zhao@northwestern.edu

 

PONE-D-19-22307

A network analysis revealed the essential and common downstream proteins related to inguinal hernia

PLOS ONE

Dear Dr. Zhao,

Thank you for submitting your manuscript to PLOS ONE. After careful consideration, we feel that it has merit but does not fully meet PLOS ONE’s publication criteria as it currently stands. Therefore, we invite you to submit a revised version of the manuscript that addresses the points raised during the review process.

The article is potentially interesting but some issues should be fixed before we can reconsider ti.

We would appreciate receiving your revised manuscript by Dec 02 2019 11:59PM. To enhance the reproducibility of your results, we recommend that if applicable you deposit your laboratory protocols in protocols.io, where a protocol can be assigned its own identifier (DOI) such that it can be cited independently in the future. For instructions see: http://journals.plos.org/plosone/s/submission-guidelines#loc-laboratory-protocols

• A rebuttal letter that responds to each point raised by the academic editor and reviewer(s). This letter should be uploaded as separate file and labeled 'Response to Reviewers'.

• A marked-up copy of your manuscript that highlights changes made to the original version. This file should be uploaded as separate file and labeled 'Revised Manuscript with Track Changes'.

• An unmarked version of your revised paper without tracked changes. This file should be uploaded as separate file and labeled 'Manuscript'.

We look forward to receiving your revised manuscript.

Kind regards,

Prof. Raffaele Serra, M.D., Ph.D

Academic Editor

PLOS ONE

Journal Requirements:

Thank you for these useful links to help us improve the manuscript. We have revised our manuscript to meet PLOS ONE's style requirements such as revising the style of the title page, including page and line numbers, adjusting heading font size, modifying reference style, and inserting figure captions and tables immediately after the first paragraph in which the figures and tables are cited.

We have added the caption for the table and the one-line title in the Supporting Information file at the end of our manuscript. All changes have been highlighted in blue (page 39, lines 726-728). Supporting information file was uploaded separately.

Additional Editor Comments:

This paper is potentially interesting for our journal, but I have some suggestions in order to improve the manuscript.

1. English language in your submission needs revision for style. It is a little hard to read. I fully understand how hard this can be, especially if English is not your first language. However, for readers to fully understand and benefit from your work it is crucial that the use of English is to a very high standard.

Thank you for your helpful comment. We have thoroughly revised the English language in this manuscript to meet the standards of PLOS ONE. All changes have been highlighted in blue in the “Revised Manuscript with Track Changes”.

2. Your Discussion Section is too superficial in the current format. You have to discuss deeply your results, and in a more critical way. In particular, can you add a future perspectives comment based on your results?

Thank you for these constructive comments. We have added a paragraph to discuss the future perspectives of our study (pages 26-27, lines 460-480). We included this paragraph here for your convenience, “Inguinal hernia formation is associated with increased lower abdominal muscle tissue fibrosis and muscle atrophy [49]. The exact role of these essential and common downstream proteins and the related signaling pathway in fibroblasts and myocytes of lower abdominal muscle tissue is unknown. Future studies will reveal the underlying molecular mechanisms for these proteins and pathways in fibroblast proliferation and fibrosis, myocyte function, and hernia formation and further target these essential and common downstream proteins for developing novel pharmacological approaches for preventing and treating recurrent inguinal hernia in high risk individuals. Our data along with previous findings regarding the effect of inhibitory mutation of PIK3R1 on SHORT syndrome-associated inguinal hernia indicate that the PI3K pathway, especially the essential protein, PIK3R1 is necessary for inguinal hernia development. A previous study showed that direct targeting of PIK3R1 in hepatic stellate cells inhibits liver fibrosis, indicating that PIK3R1 is probably participating in hernia-associated fibrosis [70]. In addition, overexpression of pik3r1 in rat myotubes reduced insulin-stimulated PI3K/AKT activation [71], and pik3r1 overexpression in mice decreased skeletal muscle insulin signaling [72]. Mice lacking both pik3r1 and pik3r2 in skeletal muscles exhibited severely impaired PI3K signaling in those muscles [73]. These animals showed reduced myocyte size and insulin-resistance in their skeletal muscles, demonstrating that in vivo class IA PI3K is both a vital regulator of muscle growth and a critical mediator of insulin signaling in the muscle. With these findings, we are planning to selectively delete PIK3R1 in fibroblasts and/or myocytes to define the relative roles of PIK3R1 in fibroblasts and myocytes for maintaining abdominal muscle function and in pathologic processes such as fibrosis, atrophy, and hernia formation. ”.

We have also discussed the clinical significance of the most significant essential protein PIK3R1 (page 26, lines 453-458). We added these two sentences to the paragraph, “ A previous study demonstrated that heterozygous mutations of PIK3R1(R649W) and the resultant impairment of the PI3K activation have been identified in patients with SHORT syndrome - a disorder characterized by Short stature, Hyperextensibility of joints and/or inguinal hernia, Ocular depression, Reiger anomaly and Teething delay [69]. This finding further emphasizes that PIK3R1 in the PI3K pathway is closely associated with the development of inguinal hernia.” 

We have fully reorganized the discussion part of this manuscript and have added the extra content to accommodate your comments (page 24, lines 406-410 and page 25, lines 426-427, lines 431-431, and lines 439-445).

Reviewers' comments:

Reviewer's Responses to Questions

Comments to the Author

1. Is the manuscript technically sound, and do the data support the conclusions?

Reviewer #1: Yes

2. Has the statistical analysis been performed appropriately and rigorously? 

Reviewer #1: Yes

3. Have the authors made all data underlying the findings in their manuscript fully available?

Reviewer #1: Yes

4. Is the manuscript presented in an intelligible fashion and written in standard English?

Reviewer #1: Yes

5. Review Comments to the Author

Reviewer #1: This is an interesting and important manuscript that adds significantly to the field and will have a noticeable impact. The manuscript is well-written and concise. The authors describe an interesting, well-thought and thoroughly designed approach for generation and analysis of PPI network associated with inguinal hernia. Using this technique, they were able not only to generate and analyze a PPI network containing proteins encoded by the inguinal hernia causative genes, but also to identify and characterize the essential and common downstream proteins related to the pathogenesis of this malady. What is even more important, this approach can be utilized for the analysis of the PPI networks associated with various human diseases. In other words, this work reports a crucial blueprint that will definitely simplify future research on bioinformatics-based identification of important players of various diseases. Furthermore, identified in this study essential and common downstream proteins related to inguinal hernia can serve as potential targets in the development of future therapeutics.

Thank you very much for the review of our manuscript and for your extremely positive comments. We have incorporated your comments into our discussion to speculate the future direction of this study (page 23, lines 394-396; page 26, lines 463-467).

6. PLOS authors have the option to publish the peer review history of their article (what does this mean?). If published, this will include your full peer review and any attached files.

Do you want your identity to be public for this peer review? For information about this choice, including consent withdrawal, please see our Privacy Policy.

Reviewer #1: No

---

## [Decision Letter · Decision Letter 1]

10 Dec 2019

A network analysis revealed the essential and common downstream proteins related to inguinal hernia

PONE-D-19-22307R1

Dear Dr. Zhao,

We are pleased to inform you that your manuscript has been judged scientifically suitable for publication and will be formally accepted for publication once it complies with all outstanding technical requirements.

With kind regards,

Prof. Raffaele Serra, M.D., Ph.D

Academic Editor

PLOS ONE

Additional Editor Comments (optional):

amended manuscript is acceptable

Reviewers' comments:

Reviewer's Responses to Questions

**Comments to the Author**

1. If the authors have adequately addressed your comments raised in a previous round of review and you feel that this manuscript is now acceptable for publication, you may indicate that here to bypass the “Comments to the Author” section, enter your conflict of interest statement in the “Confidential to Editor” section, and submit your "Accept" recommendation.

Reviewer #1: All comments have been addressed

2. Is the manuscript technically sound, and do the data support the conclusions?

Reviewer #1: Yes

3. Has the statistical analysis been performed appropriately and rigorously? 

Reviewer #1: Yes

4. Have the authors made all data underlying the findings in their manuscript fully available?

Reviewer #1: Yes

5. Is the manuscript presented in an intelligible fashion and written in standard English?

Reviewer #1: Yes

6. Review Comments to the Author

Reviewer #1: All comments, recommendations, and critiques were adequately addressed and the manuscript was revised accordingly.

7. PLOS authors have the option to publish the peer review history of their article (what does this mean?). If published, this will include your full peer review and any attached files.

Reviewer #1: Yes: Vladimir Uversky

---

## [Editor Report · Acceptance letter]

20 Dec 2019

PONE-D-19-22307R1 

A network analysis revealed the essential and common downstream proteins related to inguinal hernia 

Dear Dr. Zhao:

I am pleased to inform you that your manuscript has been deemed suitable for publication in PLOS ONE. Congratulations! Your manuscript is now with our production department. 

With kind regards,

on behalf of

Prof. Raffaele Serra 

Academic Editor

PLOS ONE